# Psychosocial Determinants of Loneliness in the Era of the COVID-19 Pandemic—Cross-Sectional Study

**DOI:** 10.3390/ijerph191911935

**Published:** 2022-09-21

**Authors:** Matylda Sierakowska, Halina Doroszkiewicz

**Affiliations:** 1Department of Integrated Medical Care, Faculty of Health Sciences, Medical University of Bialystok, 15-096 Bialystok, Poland; 2Department of Geriatrics, Faculty of Health Sciences, Medical University of Bialystok, 15-471 Bialystok, Poland

**Keywords:** loneliness, COVID-19 pandemic, mental health, quality of life

## Abstract

Background: The COVID-19 pandemic affected the mental health and social behavior of people around the world. Due to epidemiological restrictions, the period of forced isolation contributed to the feeling of loneliness. The aim of the research is to identify factors and conditions associated to the feeling of loneliness in the era of the COVID-19 pandemic. Methods: The survey was conducted among 262 people from the north-eastern Polish area, using an online survey. The diagnostic survey method was used, using the De Jong Gierveld Loneliness Measurement Scale, the Generalized Self-Efficacy Scale (GSES), the WHOQoL-Bref questionnaire. Results: A statistically significant relationship was observed between the feeling of loneliness and areas of quality of life, especially psychological and social, generalized self-efficacy and marital status and way of living (*p* < 0.05). Higher levels of stress, social distancing, restrictions at work, health status were significantly correlated with an increase in loneliness. Remote work was associated with a lower assessment of the quality of life in the psychological field (*p* < 0.05). Conclusions: Higher levels of loneliness were significantly more likely to affect people living alone and not in a relationship. Higher levels of loneliness were significantly associated with lower quality of life in the social and psychological domains, lower levels of self-efficacy, and remote work.

## 1. Introduction

The COVID-19 pandemic has affected the mental health and social behavior of people around the world [1,2,3]. As a result of the introduced restrictions in the living and economic space (quarantine, social distancing, learning and remote work), many institutions have suspended their activities. Access to basic services, including medical services, has been significantly restricted. In many places, the traditional support system has ceased to function. Due to the “novelty” of this situation, manifested in, among others, impoverishment of the possibility of contact with other people, difficulties in participating in social or professional life, various manifestations of people’s reactions in such a difficult time were observed [4,5,6].

In the past, many studies have been carried out to assess psychological problems resulting from previous pandemics. The literature reports that containment activities such as quarantine, isolation, and social distancing have had an impact on people’s mental well-being, as well as on emotional responses to the pandemic itself, which manifested themselves in non-adaptive behaviors, emotional anxiety, and defensive responses: fear; frustration; loneliness; anger; boredom; depression; stress; avoidance calculations [7,8,9]. The COVID-19 pandemic has also been linked to the health, both somatic and mental, of the population. In many countries (e.g., Italy), strict social distancing and sanitary measures have been effective in reducing new infections [10]. They also had immediate and unprecedented consequences for the psychosocial functioning of individuals. Studies have shown that social distancing orders and lockdowns have caused significant disruption to people’s behaviors and daily habits, leading to social isolation and negative repercussions on well-being and mental health [11,12]. Other reports indicate that the feeling of uncertainty and danger due to the pandemic, the loss of professional and economic stability, resulted in a lack of control over one’s own life, increasing stress and reducing the quality of life [13,14]. Not without influence on contemporary reactions were information and social media, providing constantly, from many sources, disturbing information from more or less scientific sources [7,10].

The period of isolation in homes during the COVID-19 pandemic has also contributed to the feeling of anxiety about health and future life, depressive moods, intensifying the feeling of loneliness observed in different age groups, with greater severity in the elderly [15,16]. Social isolation and loneliness often coexist with each other. Social isolation means having a small social network and little interaction with others. It is defined by the level and frequency of social contacts. It is a risk factor for the development of loneliness, despite the fact that some people like this state, on the other hand, maintaining social relationships does not guarantee that loneliness will not develop [17].

Loneliness, defined as a subjective, emotional state of feeling social isolation and a sense of being cut off from others, is a timeless and cross-cultural phenomenon, present in the life of every person. It arises when the desire for social relationships is insufficient, quantitatively or qualitatively [18,19]. Cacioppo and Patrick argue that the mere presence of others does not make people feel less alone; rather, they need the presence of someone they trust and can share common goals, plan for the future, and work together to survive and prosper [20]. Theory and empirical evidence suggest two basic dimensions: social loneliness and emotional loneliness. According to Weiss, social loneliness stems from an unmet need for social peer relationships and is thus experienced by socially poorly integrated individuals, while emotional loneliness results from an unmet need for close, intimate, or emotional contact with significant people, such as a partner, parents, or children [21]. Research indicates that loneliness and social isolation were common in Europe, the U.S. and China (10–40%) even before the pandemic and were described as a ‘behavioral epidemic’ The situation has worsened due to restrictions imposed by COVID-19 [22,23,24]. About a quarter (24%) of community-dwelling Americans aged 65 and older report feelings of social isolation, and 35% of adults aged 45 and over and 43% of those aged 60 and over report feeling lonely [25]. Social ties have been clearly disrupted during the pandemic, making people more lonely.

Understanding loneliness is important due to its relationship to the mental and physical health of the individual. Researchers show that social isolation and feelings of loneliness lead to feelings of constant stress and depression, and also trigger physiological changes, linked to the immune system and inflammatory responses, that can further exacerbate health [19,26,27]. As a result of a persistent sense of loneliness, susceptibility to mental disorders and psychosomatic diseases may also increase [28]. Loneliness and social isolation are therefore important determinants of health and are associated with quality of life measures and mental disorders such as depression, as well as physical diseases such as cardiovascular disease and hypertension [29,30]. Loneliness also translates into social functioning. A long-term state of loneliness develops negative attitudes, such as, for example, shyness, low mood, fear of judgment, lower social skills. Greater negative social expectations in single people can lead to the development of distrust, hostility and intolerance. Single people are much more sensitive to threats and have less ability to adapt and respond to stress [31].

In the era of the COVID-19 pandemic, loneliness, due to the ongoing pandemic restrictions, has become an everyday reality [31]. Given the likelihood of future waves of epidemics and related restrictions, public health should prioritize addressing the root causes of loneliness and social isolation, and in particular addressing the needs of specific social groups, particularly those who are vulnerable, stress-prone, or single. Greater emphasis should be placed on primary prevention and population strategies to promote mental health. Bearing in mind that the effects of loneliness can be long-term, it seems reasonable to conduct research in order to recognize and understand this phenomenon, which could be a prerequisite for taking adequate corrective and in the future preventive actions.

The aim of the research is to identify factors and conditions related to the feeling of loneliness in the era of the COVID-19 pandemic.

Answers to the following questions are sought:What is the level of loneliness in the study group and what are the links between sociodemographic and psychosocial factors?Is there a link between feelings of loneliness and the quality of life of the subjects?Is there a link between feelings of loneliness, self-efficacy and quality of life and the way you work during the ongoing pandemic?

## 2. Material and Methods

### 2.1. Research Material

The research was addressed to people from the area of north-eastern Poland (Podlaskie Voivodeship). A total of 262 people took part in the study.

The cross-sectional study was conducted during the COVID-19 pandemic, from November 2021 to March 2022. The survey was conducted using an online survey created in the Google online platform.

An invitation to participate in the study was sent to adults via social media. All respondents were Poles. Inclusion criteria: age over 18 years. Respondents’ responses were recorded on the Google platform they were using. The raw data, saved in Excel, was downloaded for statistical analysis.

Participation in the study was voluntary. The studies were anonymous, any participant could withdraw from the study at any time. Joining the study was tantamount to agreeing to participate in the study.

### 2.2. Research Methods

The research used the diagnostic survey method, using the following research tools:

*Questionnaire of own construction questionnaire* (17 questions), containing imprint data (10 questions) and referring to: subjective assessment of the level of stress (assessed on the Likert scale from 1 to 5), the impact of the pandemic on the reduction in work (1–5), social distancing (1–5), health assessment (1–5), lack of emotional control, aggressive behavior (1–5) and experiencing negative emotions (1–5), where 1—is the lowest value and 5—is the highest value.

*Generalized Self-Efficacy Scale (GSES)* according to R. Schwarzer, M. Jerusalem, Z. Juczyński, is a research tool, consisting of 10 questions, designed to study the general belief of an individual about effectiveness in dealing with emerging difficulties and obstacles. The internal consistency by Cronbach’s alpha for the scale is 0.85. The study participant responds to the claims on a 4-point scale. The sum of all points gives an overall indicator from 10 to 40. The general indicator, when converted into standardized units, shall be interpreted. Results in the range of 1–4 sten were assumed to be treated as low, 5–6 sten as average, and 7–10 sten as high [32].

*De Jong Gierveld’s Scale for Measuring the Feeling of Loneliness (DJGLS)*, in Polish adaptation according to P. Grygiel, G. Humenna, S. Rębisz, P. Świtaj, J. Sikorska. The tool consists of 11 statements, of which 6 items contain negatively formulated sentences, describing the lack of satisfaction with social contacts, and the remaining 5—positively formulated—measuring satisfaction related to interpersonal relationships. The internal consistency by Cronbach’s alpha for the scale is 0.80.

The loneliness index (11–55) is calculated by recoding the “negative” items and then adding up all the test items [33].

*WHOQoL-Bref—*questionnaire for assessing the quality of life—is a tool designed to assess the overall quality of life of both healthy and ill people. It was created on the basis of the WHOQOL-100 scale. The abbreviated version of the indicator consists of 26 questions and allows you to assess the quality of life in four areas: physical; psychological; social; and environmental. The internal consistency for the total scale is 0.90. The internal consistency of the domains, assessed by Cronbach’s alpha, included the following; physical; psychological; social relationships; and environmental, 0.81, 0.78, 0.69, and 0.77, respectively.The score for the domains is determined by calculating the arithmetic mean from the items included in the individual domains (4–20). The scoring of fields has a positive direction, as more points mean a better quality of life [34].

The research was carried out in accordance with the Declaration of Helsinki and Good Clinical Practice in research. Bioethics Committee of the Medical University in Bialystok, Poland granted the ethical approval for the study (APK.002.292.2021, APK.002.80.2022). Participation was voluntary and participants were informed about the project.

### 2.3. Statistical Analysis

The obtained results were subjected to statistical analysis, in which the arithmetic mean and standard deviation were calculated as well as the values of both minimum and maximum quantitative variables, while the percentage distribution was calculated for qualitative variables.

The relationships between pairs of quantitative variables were analyzed based on the Spearman rank correlation coefficient. For nominal characteristics, the significance of differences between groups was assessed using the Mann–Whitney U test or the Kruskal–Wallis U test (for more than two groups).

The *p* < 0.05 level was assumed as a statistically significant relationship (*); *p* < 0.01 is a highly significant relationship (**); *p* < 0.001 is a very highly statistically significant relationship (***).

The data was processed in a Microsoft Excel 2013 spreadsheet and analyzed using Statistica v.13, StatSoft, Poland. 

## 3. Results

### 3.1. Socio-Demographic Characteristics of the Surveyed Persons

The study included 262 people aged 18 to 84 years. The average age of the subjects was, respectively—45.8 years, the youngest was at the age of 18 years, and the oldest—84 years. More than 80.0% of people were women, in a similar percentage (82.1%) people living in the city. More than three-quarters of the respondents (75.6%) had a university degree. An analysis of their marital status showed that almost three-quarters of the respondents (71.4%) were in a relationship. In the study group, the vast majority (65.3%) were economically active people (Table 1).

Subsequently, the survey was interesting to get to know the opinions of the respondents on possible changes in the mode of performing the current work, in connection with the restrictions of the COVID-19 pandemic.

As the analyses showed, as many as every second person (49.3%) was forced to switch to the remote work/learning mode, while 40.0% of respondents declared that during the pandemic they did not change their current form of professional activity, and 7.0% limited/suspended their work. According to the results, for almost a third of respondents (29.5%), the pandemic period significantly limited earning opportunities.

Subsequently, the study analyzed the level of perceived stress in the study group. More than a third (37.4%) indicated very high/high levels of stress, as did 34.0%—medium and 28.6%—light.

For nearly two-thirds of respondents (63.8%), the pandemic period has significantly reduced social contacts. Only one in ten people (12.2%) indicated a low degree of social distancing.

Next, the health status of the subjects was analyzed. As the results showed, every second person (55.0%) rated their health as very good/good, a third (34.7%) as average, every tenth (10.3%) as bad. More than 40.0% indicated that they suffer from chronic diseases. Almost every third person (28.2%) significantly experienced physical impairments due to the disease. Detailed data are presented in Table 1.

### 3.2. Level of Loneliness, Quality of Life and Generalized Self-Efficacy in the Study Group

The average score of loneliness, assessed using the DJGLS scale (11–55), in the study group was, respectively—26.7 ± SD 8.4, which generally indicates an average severity of loneliness.

The average scores in individual areas of quality of life were, respectively: in the physical domain—14.7 ± 2.4 SD, in the psychological domain—12.8 ± 2.8 SD, in the field of social relations—13.9 ± 3.6 SD and in the environmental category—12.6 ± 2.5 SD. Analyses showed that the respondents obtained the lowest score values in the fields of environment and psychology.

The average score of generalized self-efficacy according to the GSES scale was—29.4 ± 4.1 SD, which in the transformation into units is standardized at the level of just over 6 sten (average self-efficacy). Detailed data are presented in Table 2.

### 3.3. Feeling Lonely, Linked to Quality of Life and Generalized Self-Efficacy

Subsequently, the study was interesting to determine whether there is a relationship between the feeling of loneliness and individual areas of quality of life and generalized self-efficacy.

A statistically significant relationship was observed between the feeling of loneliness and the studied areas of quality of life, especially the psychological and social domain (*p* = 0.0000).

It was observed that feelings of loneliness were also significantly correlated with generalized self-efficacy (*p* = 0.0000) (Table 3).

The dependencies of the sense of loneliness in connection with the quality of life and generalized self-efficacy are illustrated in the scatter diagrams below (Figure 1).

### 3.4. The Level of Loneliness in Relation to Sociodemographic Factors

Subsequently, the study analyzed the impact of selected sociodemographic factors (gender, age, social contacts) and health status on the level of loneliness (Table 4).

In the course of the analysis, no statistically significant differences were found between the level of loneliness and gender, place of residence, education, having children, chronic disease (*p* value > 0.05). On the other hand, statistically significant relationships were found between marital status and living with a close person and a sense of loneliness (*p* < 0.05). Detailed data are presented in Table 4.

No statistically significant relationships were found between age and the feeling of loneliness (*p* = 0.2201). Spearman’s correlation coefficient between age and loneliness was at a very low level (*r*_S_ = −0.08) (Figure 2).

The analysis in age groups shows a statistically significant difference in the sense of loneliness, which is the highest among people aged under 35 and 56+, and the lowest among middle-aged people. Despite the statistical significance, these differences are small (data in Table 5).

### 3.5. Feelings of Loneliness in Connection with Psychosocial Factors

Subsequently, analyses were made between factors such as: stress level; social distancing; work limitation; assessment of the health and mental condition of the surveyed people; and the feeling of loneliness.

Analyses showed that a higher sense of social distancing, the experience of negative emotions and higher levels of stress, were statistically significantly linked to an increase in feelings of loneliness. In addition, the results showed that a higher health score was associated with lower levels of loneliness (*r*_S_ = −0.33) (Table 6).

The obtained results, presented below on box-plot charts, present the average level of loneliness, relative to the assessments made for individual spheres of life along with the range of 95% confidence interval and the typical variability interval (Figure 3).

### 3.6. Feeling Lonely, Self-Efficacy and Quality of Life in Connection with the Mode of Work Performed

The results of the analyses showed that more than half (53.2%) of the respondents changed their work to remote mode during the pandemic. The analyses tried to determine whether performing work remotely was related to a sense of loneliness, generalized self-efficacy and quality of life.

The results showed a relationship was observed between the mode of work performed and the subjective assessment of the quality of life in the psychological domain. A higher quality of life was presented by people whose work performance had not changed (Table 7).

## 4. Discussion

The aim of this study was to identify factors and conditions related to the feeling of loneliness in the era of the COVID-19 pandemic. Our results indicated that factors related to loneliness during the pandemic were: feeling under stress; limiting social contacts and limiting work; worse subjective assessment of health; and experiencing negative emotions.

Previous research conducted among populations affected by the COVID-19 pandemic clearly indicates its impact on the mental condition of society, which may be caused by restrictions, social restrictions, as well as a sense of helplessness and powerlessness in the fight against the coronavirus. To a very large extent, the possibility of direct interpersonal contacts, which, according to specialists, are necessary to maintain the psychological balance of a person, has been limited [35,36].

The results obtained in their own work showed that the feeling of loneliness of the subjects, associated with the COVID-19 pandemic, was associated with a lower quality of life in the social and psychological field. Psychologists emphasize that the greatest importance, both for loneliness and life satisfaction, is the assessment of social contacts, their capabilities and limitations. This is probably due to the fact that contacts with other people determine the individual’s belonging to different social groups and are a determinant of the social networks, to which the individual belongs. In this way, the need for affiliation can be met, which results in a better assessment of the quality of life and satisfaction with it. On the other hand, during the deprivation of this need, a feeling of loneliness may occur [35,36]. The possibility of social contacts has an undeniable relationship with the perceived satisfaction or loneliness. If there are no limits, you can simultaneously get social support, which is one of the predictors of life satisfaction [37].

In their own survey, for nearly two-thirds of respondents (>63%), the pandemic period significantly reduced social contacts. It was also observed that people who felt a greater intensity of loneliness reported lower self-efficacy. Other studies also have found significant links between self-efficacy and the mental health and well-being of the individual [38].

According to Gu et al., the stronger an individual’s loneliness, the higher the person’s levels of anxiety, depression, and stress, and the worse the level of mental health [39]. Previous research, even before the COVID-19 pandemic, showed that when individuals are in a state of loneliness, their mental resilience and ability to regulate emotionally decreases, and their coping styles change in the face of external events, often adopting negative and non-adaptive coping styles [40,41,42]. The actual research also indicated that the feeling of stress for the majority of respondents was associated with a greater sense of loneliness.

The results of the analysis of Chinese studies conducted during the COVID-19 pandemic in 2021 confirmed that loneliness was significantly positively correlated with the levels of anxiety, depression and stress [35]. According to other authors, the COVID-19 outbreak has severely disrupted physical activity, sleep quality, and mental health. Negative social emotions, such as worry, anxiety, depression, and stress, fluctuated at high levels [43,44].

The literature reports indicate that women are significantly more likely than men to show tendencies to the occurrence of anxiety, depressive disorders and experiencing negative emotions [44,45,46,47,48]. Other researchers also argue that the experience of loneliness during the COVID-19 pandemic is associated with several risk factors, including those related to the female sex [33,49]. In the results obtained in the presented work, those regarding mental health, are not representative in relation to the overall population assessment, due to the significant predominance of women (>80%). The results obtained in the presented study concerning the feeling of loneliness did not show any significant correlation with gender.

The analysis in age groups indicated a statistically significant difference in the sense of loneliness, which is the highest among people aged under 35 and ≥56 years, and the lowest among middle-aged people. This is confirmed by the other authors (Perlman and Landolt) [50] who indicate complex relationships occurring between the phases of life and loneliness. Loneliness is high among the youngest respondents, followed by a period of decreased loneliness in the middle age, rising again among the older adults.

The results obtained in the course of their own work were also of interest, indicating that loneliness, felt during the pandemic, was more relevant to people without relationships, living alone. According to Lampraki (et al.), the pandemic has led to an increased need for closeness with significant people such as parents, emotional partners or children, perhaps due to the stress experienced in the face of the threat of a pandemic, which has resulted in an increase in emotional loneliness, caused by social distancing [51]. This seems to confirm that one of the factors protecting against loneliness is the presence of other people. In addition, numerous authors focus on studying the predictors of loneliness in the form of having a loved one, a partner (emotional loneliness), and the social network to which a person belongs (social loneliness) [37].

Research by Casale et al. also indicates that stressful situations increase the need for the individual, especially those very concerned about their interpersonal needs in a situation of social isolation, to support and interact with others [49]. Losada-Baltar et al., on the basis of a Spanish sample, confirmed the above hypothesis, stating that people during the current pandemic felt less lonely when they lived with others and had more contact with relatives [52]. The literature reports also indicate that support is associated with improved health by reducing loneliness [39,53].

Of the respondents presented in our own work (mostly people with higher education), almost 60% declared that the pandemic did not affect the limitation of their earning opportunities. On the other hand, remote work, affecting almost half of the respondents, due to its limitations in direct social contacts, was associated with a lower, subjective assessment of the quality of life in the psychological field.

It is worth noting here that one of the criteria for better professional adaptation in the face of the crisis is the ability to perform professional duties remotely, which, however, are available for selected professions, usually with higher qualifications. This also raises new challenges, including those related to proper work management, monitoring and confidence [54,55].

Referring to health status, our research also found that an inferior, subjective assessment of physical health correlates with a higher sense of loneliness. This is confirmed by Pai and Vella’s research, indicating that people with lower subjective health suffer from loneliness as the disease consequently weakens their social ties [51]. Other, global studies on a large population indicate several factors related to loneliness, minimum financial insecurity and poor physical and mental health [56].

### Limitations Study

The presented studies have their limitations, which should be taken into account when interpreting the results. First of all, these are cross-sectional studies, based solely on the study of self-assessment. Although the standardized research tools used in this study are sensitive instruments designed to detect different states and characteristics, all responses focus on the subjective feelings of the respondents rather than objective criteria, which creates the risk of false positive results. Furthermore, the research sample (n = 262) is not representative, it is the group of people who joined the research. Therefore, the results cannot be generalized to the entire Polish population. Secondly, the study was conducted online, which meant that the researchers had no influence on the representation of the group (participation was voluntary, people interested in the topic undertook to complete the survey in the Google form). This resulted in a much greater representation of people with higher education, which makes it difficult to relate the results of the research to the general population. In future research, we would like to broaden the analysis to look for the relationship between other variables related to pandemic limitations and mental health.

In addition, the level of loneliness, self-efficacy and quality of life from before the pandemic were not taken into account, which may have been relevant to the current level of these variables. Nevertheless, our study provides solid and consistent evidence that feelings of loneliness are a significant problem during the pandemic that should not be ignored. These aspects should be recognized, considered and properly taken into account in psychological interventions that counteract the risk of mental disorders related to the pandemic. It should be borne in mind that the pandemic period is a special time for everyone, which is the reason for a need of joint actions of decision-makers, health care workers, science and education centers and law enforcement services to lead to the alleviation of the feeling of uncertainty, stress and fears of society, and as a result, improvement in the quality of life.

## 5. Conclusions

Our research showed that the increase in the level of loneliness was closely related to the marital status of the subjects and the way of living. Significantly, more often it concerned people who were not in a relationship and living alone. Among psychosocial factors, the increase in the level of loneliness was significantly influenced by: higher levels of perceived stress; experiencing negative emotions; limitations of social contacts and work performance; and worse assessment of health. The increase in feelings of loneliness was correlated with lower quality of life of the subjects in the social and psychological field and lower levels of self-efficacy in coping with difficulties. People working remotely showed a lower quality of life in the psychological field.

## Figures and Tables

**Figure 1 ijerph-19-11935-f001:**
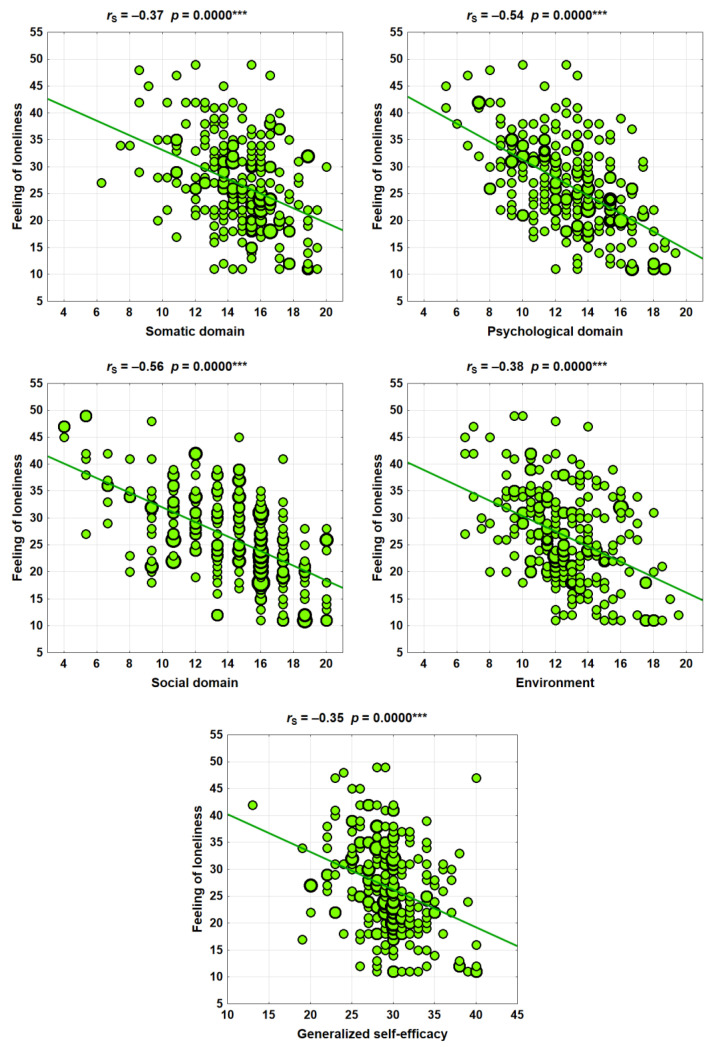
Feeling lonely, linked to quality of life in four areas: physical; psychological; social and environmental; and generalized self-efficacy. *p* < 0.001 is a very highly statistically significant relationship (***).

**Figure 2 ijerph-19-11935-f002:**
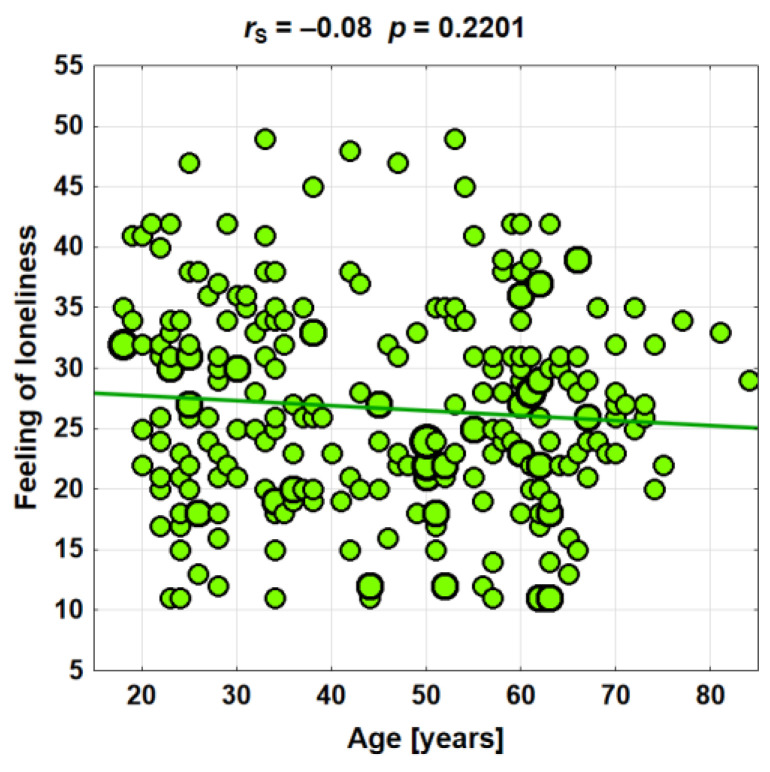
A sense of loneliness in connection with age.

**Figure 3 ijerph-19-11935-f003:**
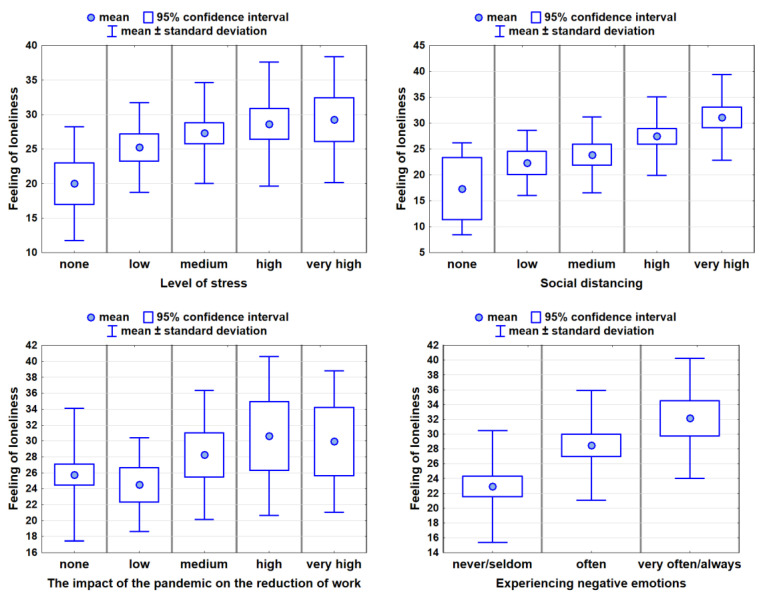
Feelings of loneliness in connection with psychosocial factors.

**Table 1 ijerph-19-11935-t001:** Characteristics of the subjects (n = 262).

Variables	Total n = 262
Age in years,	
Mean ± SD	45.8 ± 16.6
Min.	18
Max.	84
Median (Q1–Q3)	47.5 (30–60)
Gender, (%)	
-women	80.9
-men	19.1
Education, (%)	
-vocational	3.0
-medium	21.4
-higher	75.6
Place of residence, (%)	
-city	82.1
-village	17.9
Marital status, (%)	
-in a relationship	71.4
-single person	28.6
Having children, (%)	
-yes	33.2
Social status, (%)	
-working person	65.3
-pensioner	24.0
-student	9.6
-unemployed	1.1
Living with a loved one (%)	
-yes	85.5
Change in activity professional during the pandemic, (%)	
-no change	43.7
-work/remote learning	49.3
-restriction or suspension of work	7.0
Impact of the pandemic to limit the possibilities earnings, (%)	
-very large/large	16.1
-medium/to a small extent	25.2
-no impact	58.7
The level of stress felt, (%)	
-very high/high	37.4
-medium	34.0
-light/none	28.6
Degree of limitation social contacts, (%)	
-very	26.0
-quite significantly/medium	57.6
-to a small extent/no impact	16.4
Self-assessment of health, (%)	
-very good/good	55.0
-average	34.7
-very bad/bad	10.3
Does he have a chronic disease?	
-yes (%)	40.8
Degree of limitation of mobility in connection with the disease, (%)	
-very high/high	14.0
-medium	11.2
-small/none	74.8

Abbreviations: SD—standard deviation; Q1—lower quartile; Q3—upper quartile, Min.—minimum; Max.—maximum.

**Table 2 ijerph-19-11935-t002:** Average scores of loneliness, quality of life, and generalized self-efficacy of subjects (n = 262).

Variables	$\bar{x}$ (95%)	Me	SD	Q1	Q3	Min	Max
WHOQoL-BREF (0–20)							
Somatic, mean ± SD	14.7 (14.4–15.0)	14.9	2.4	13.1	16.0	6.3	20.0
Psychological field, mean ± SD	12.8 (12.4–13.1)	12.7	2.8	10.7	14.7	5.3	19.3
Social field, mean ± SD	13.9 (13.5–14.3)	14.7	3.6	12.0	16.0	4.0	20.0
Environment, mean ± SD	12.6 (12.3–12.9)	12.5	2.5	11.0	14.0	6.5	19.5
Feeling lonely (PSS-10), mean ± SD (11–55)	26.7 (25.7–27.7)	26	8.4	21	32	11	49
Self-efficacy (GSES), mean ± SD (10–40)	29.4 (28.9–29.9)	30	4.1	27	31	13	40

**Table 3 ijerph-19-11935-t003:** A sense of loneliness in correlation with quality of life and generalized self-efficacy.

Variables	Feeling of Loneliness
WHOQoL-BREF	
Somatic domain	−0.37 (*p* = 0.0000 ***)
Psychological domain	−0.54 (*p* = 0.0000 ***)
Social domain	−0.56 (*p* = 0.0000 ***)
Environment	−0.38 (*p* = 0.000 ***)
GSES	−0.35 (*p* = 0.0000 ***)

*p* < 0.001 is a very highly statistically significant relationship (***).

**Table 4 ijerph-19-11935-t004:** A sense of loneliness in connection with sociodemographic factors.

**Variables**	**Feeling of Loneliness (*p* = 0.4887)**					
**Sex**	**N**	** $\bar{x}$ **	**Me**	**SD**	**Min**	**Max**
woman	212	26.8	26.5	8.2	11	49
man	50	26.2	24.5	9.1	11	49
**Place of Residence**	**Feeling of Loneliness (*p* = 0.5806)**					
	**N**	** $\bar{x}$ **	**Me**	**SD**	**Min**	**Max**
village	47	27.8	28	7.8	15	47
city up to 100 thousand residents	45	27.1	26	8.8	11	49
city over 100 thousand residents	170	26.3	26	8.4	11	49
**Education**	**Feeling of Loneliness (*p* = 0.4357)**					
	**N**	** $\bar{x}$ **	**Me**	**SD**	**Min**	**Max**
other	64	27.4	27	8.2	11	45
higher	198	26.5	26	8.5	11	49
**Marital Status**	**Feeling of Loneliness (*p* = 0.0090 **)**					
	**N**	** $\bar{x}$ **	**Me**	**SD**	**Min**	**Max**
In a relationship	187	25.8	25	8.3	11	49
Single person	75	28.8	29	8.3	12	49
**Children**	**Feeling of Loneliness (*p* = 0.6227)**					
	**N**	** $\bar{x}$ **	**Me**	**SD**	**Min**	**Max**
yes	87	26.5	25	9.3	11	49
no	175	26.7	26	7.9	11	49
**Living with a Significant Other**	**Feeling of Loneliness (*p* = 0.0088 **)**					
	**N**	** $\bar{x}$ **	**Me**	**SD**	**Min**	**Max**
yes	224	26.1	25	8.4	11	49
no	38	29.9	30	7.8	11	49
**Chronic Disease**	**Feeling of Loneliness (*p* = 0.3030)**					
	**N**	** $\bar{x}$ **	**Me**	**SD**	**Min**	**Max**
yes	155	26.2	26	8.6	11	49
no	107	27.3	27	8.1	11	49

*p* < 0.01 is a highly significant relationship (**).

**Table 5 ijerph-19-11935-t005:** A sense of loneliness in connection with age groups.

Age [in Years] N; %	Sense of Lonliness (*p* = 0.0119 *)		
	$\bar{x}$	Me	SD
<35 (90; 34.4)	28.5	30	8.6
35–54 (87; 33.2)	25.0	23	8.5
≥55 (85; 32.4)	26.6	27	7.9

*p* < 0.05 is a statistically significant relationship (*); *p*—test probability values calculated using the Kruskal–Wallis test.

**Table 6 ijerph-19-11935-t006:** A sense of loneliness in connection with psychosocial factors.

Assessement of Life Domains (1–5)	Feeling of Loneliness
Social distancing	0.40 (*p* = 0.0000 ***)
Experiencing negative emotions	0.31 (*p* = 0.0000 ***)
Level of stress	0.28 (*p* = 0.0000 ***)
Lack of emotional control, aggressive behavior	0.18 (*p* = 0.0040 **)
The impact of the pandemic on the reduction in work	0.16 (*p* = 0.0092 **)
Health assessment	−0.33 (*p* = 0.0000 ***)

*p* < 0.01 is a highly significant relationship (**); *p* < 0.001 is a very highly statistically significant relationship (***).

**Table 7 ijerph-19-11935-t007:** A sense of loneliness and self-efficacy in connection with the mode of work performed.

**Mode of Work**	**Sense of Lonliness (*p* = 0.1907)**					
	**N**	$\bar{x}$	**Me**	**SD**	**Min**	**Max**
No change	103	25.9	24	8.4	11	49
Remote work	117	27.5	27	9.0	11	49
	**General Self-Efficacy (*p* = 0.0941)**					
	**N**	** $\bar{x}$ **	**Me**	**SD**	**Min**	**Max**
No change	103	30.0	30	3.7	19	40
Remote work	117	29.1	29	4.4	13	40
	Quality of life					
	**Somatic Domain (*p* = 0.1969)**					
	**N**	** $\bar{x}$ **	**Me**	**SD**	**Min**	**Max**
No change	103	15.1	15.4	2.2	8.0	20.0
Remote work	117	14.6	14.9	2.5	7.4	19.4
	**Psychological Domain (*p* = 0.0413 *)**					
	**N**	** $\bar{x}$ **	**Me**	**SD**	**Min**	**Max**
No change	103	13.3	13.3	2.8	6.0	18.7
Remote work	117	12.4	12.7	2.9	5.3	19.3
	**Social Domain (*p* = 0.3833)**					
	**N**	** $\bar{x}$ **	**Me**	**SD**	**Min**	**Max**
No change	103	14.0	14.7	3.7	4.0	20.0
Remote work	117	13.6	14.7	3.9	4.0	20.0
	**Environment (*p* = 0.4837)**					
	**N**	** $\bar{x}$ **	**Me**	**SD**	**Min**	**Max**
No change	103	12.7	12.5	2.5	6.5	19.5
Remote work	117	12.5	12.0	2.5	6.5	18.0

*p* < 0.05 is a statistically significant relationship (*).

## Data Availability

Data are available upon reasonable request.

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
