# Peer review of "Psychosocial Determinants of Loneliness in the Era of the COVID-19 Pandemic—Cross-Sectional Study"

_ijerph, 2022, doi:10.3390/ijerph191911935_

Round 1
Reviewer 1 Report
I would like to thank the authors for this research that aims to identify factors and conditions related to the feeling of loneliness in the era of the COVID-19 pandemic.
The research subject is timely, innovative, and highly interesting. It also fits the aim and scope of the journal.
The research is well designed and follows a sound scientific research method. Results and recommendations are clear and could have an impact among the community of researchers.
However, some modifications are needed in order to improve the quality of the paper.
You need to pay attention to the use of certain appropriate words like in the abstract and other parts of the research.
Pay attention to the repetitions observed in several parts of the research.
Pay attention to the appropriate use of pronouns.
More recent references are suggested to solidify your sources.
Two points in regard to the methodology. Why you limited your research to only the north eastern part of Poland?
Is the sample (262) statistically representative of the total population ? If not, you need to mention that as limitations.
Conclusion, recommendations and implications are clear and interesting.
Other minor comments are directly attached to the manuscript.

Author Response
Dear Reviewer,
Thank you very much for your valuable comments and tips concerning our manuscript. We have made effort to follow the recommendations and make the necessary corrections. We hope that our manuscript will meet editorial expectations in its current version.
We refer to the comments below.
You need to pay attention to the use of certain appropriate words like in the abstract and other parts of the research.
The suggested comments have been taken into account.
Pay attention to the repetitions observed in several parts of the research.
Repetitions in the text have been removed.
Pay attention to the appropriate use of pronouns.
The manuscript has been checked again by an English-language translator for grammatical correctness.
More recent references are suggested to solidify your sources.
Suggested references have been added to the thesis. Thank you for your literature suggestions.
Two points in regard to the methodology. Why you limited your research to only the north eastern part of Poland?
In our study, we assumed that the research sample would be people from the province inhabited by the authors of this study. This is due to the availability of social networks and objective difficulties resulting from pandemic constraints during the implementation of the research.
Is the sample (262) statistically representative of the total population ? If not, you need to mention that as limitations.
Undoubtedly, it would be interesting to extend the research to a larger region of Poland. We wrote about the lack of representativeness of the sample in the limitations of the study. The greater representation of women in the study probably results from the greater interest of women in participating in the survey.
Reviewer 2 Report
It seems interesting to me to develop the thought that social isolation and loneliness, although they often co-exist, are not equal and coinciding concepts. The pandemic forced many people into quarantine, into isolation, but this did not coincide with loneliness for everyone. It was isolation not chosen, to which some were able to respond with personal resources and alternatives to face-to-face relationships, which allowed them not to suffer from loneliness. Moreover, the reported data also do not indicate a severe situation. Perhaps it might be more interesting to understand why, despite the pandemic and stress, the majority of the sample feels they have good health (very good/good 55%, average 34.7%).
The correlations reported are within expectations; although the statistical significance is good, the magnitude of the correlations is generally relatively low. Moreover, when there is no significance, the data cannot be said to demonstrate a relationship (295, 447). The age of the sample is very large: from 18 to 84 years. By comparing more homogeneous age groups (perhaps by increasing the sample) the statistical magnitude might increase or other significant data might emerge.
The data in Table 2 are also given in full in the text and need not be repeated; while it would be useful to have some more information on the Mann-Whitney U-test and the Kruskal-Wallis U-test.
How are 'Experiencing negative emotions' and 'Lack of emotional control, aggressive behavior' assessed?
The discussion should focus on the study data and the comparison with the literature; instead, data from other studies are predominantly reported.
The bibliography should be reviewed against the journal's criteria.
Author Response
Dear Reviewer,
Thank you very much for your valuable comments and tips on our manuscript. We have made every effort to follow the recommendations and make the necessary corrections. We hope that our manuscript in its current version will meet editorial expectations.
We refer to the comments below.
It seems interesting to me to develop the thought that social isolation and loneliness, although they often co-exist, are not equal and coinciding concepts. The pandemic forced many people into quarantine, into isolation, but this did not coincide with loneliness for everyone. It was isolation not chosen, to which some were able to respond with personal resources and alternatives to face-to-face relationships, which allowed them not to suffer from loneliness. Moreover, the reported data also do not indicate a severe situation. Perhaps it might be more interesting to understand why, despite the pandemic and stress, the majority of the sample feels they have good health (very good/good 55%, average 34.7%).
The above data, cited by the reviewer, refer to the state of health in the physical sphere. On the other hand, the main interest of the researchers was the assessment of mental health, which we have subjected to a broader analysis.
The correlations reported are within expectations; although the statistical significance is good, the magnitude of the correlations is generally relatively low. Moreover, when there is no significance, the data cannot be said to demonstrate a relationship (295, 447).
The data that were not statistically significant were removed (295) and corrected in the conclusions (447) as suggested.
The age of the sample is very large: from 18 to 84 years. By comparing more homogeneous age groups (perhaps by increasing the sample) the statistical magnitude might increase or other significant data might emerge.
We agree with the Reviewer's opinion. In the conducted analyzes, we focused only on age, as one of the many variables potentially influencing the level of loneliness. In the future, we would like to broaden the analysis by age group
The data in Table 2 are also given in full in the text and need not be repeated; while it would be useful to have some more information on the Mann-Whitney U-test and the Kruskal-Wallis U-test.
Table 2 contains the obtained basic descriptive statistics (presents the general results of the questionnaires used in the study). In the further part of the work, statistical tests were used to show the relationships between the studied variables.
How are 'Experiencing negative emotions' and 'Lack of emotional control, aggressive behavior' assessed?
'Experiencing negative emotions' and 'Lack of emotional control, aggressive behavior' were assessed on the Likert scale (1-5). In the description of the methodology, these variables are described generally as the subjective assessment of mental health.
The discussion should focus on the study data and the comparison with the literature; instead, data from other studies are predominantly reported.
The discussions were revisited and the suggested changes were made.
The bibliography should be reviewed against the journal's criteria.
Comments were taken into account.
Reviewer 3 Report
I begin by congratulating the authors and researchers of the manuscript intitled “Psychosocial determinants of loneliness in the era of the COVID-19 pandemic –cross-sectional study”. The topic is actual and very interesting.
Personally, I disagree with the actual title. As suggestion, something like “The psychosocial determinants of Polish loneliness in the era of the COVID-19” could be more obviously informative and clear.
Please, rewrite the abstract. Remove the sentences “1)Background:” “2) Methods:” “3) Results:” “4. Conclusions:”. It’s not necessary to relate herein the statistical results (e.g. rs =-0.54, p=0.0000). And look also for more keywords.
The table 1 could be simpler. Too much information kills information.
The title of Figure 1 is currently incomplete.
Throughout the text, please reformulate the paragraphs. You make paragraphs with a single sentence. This is not at all desirable as it causes breaks in reading and interpretation. Review all the paragraphs.
You should also reformulate the section “2.2. Research Methods”. In specific, from line 140 to 146.
Line 125: Google Forms is a online platform and not a software.
About the results, I am very frustrated. You only report descriptive data. Why don’t you test any t student test in order to clearly express and show any significant difference? By instance, with “no change” and “remote work” of data from table 6.
The Discussion section is a mess in its own right. Why reintroduce literature again instead of going straight to the key points in which the results obtained from previous studies are discussed?
Arguments from line 338 and 339 are confused.
Line 357: you claim “my own research”, why? It should be “this study shows…”
In the “conclusions”, you managed to make 3 chapters with 4 sentences. Also rewrite the future research part, and add a chapter on limitations.
Good luck!
Author Response
Dear Reviewer,
Thank you very much for your valuable comments and tips on our manuscript. We have made every effort to follow the recommendations and make the necessary corrections. We hope that our manuscript will meet editorial expectations in its current version.
We refer to the comments below.
Personally, I disagree with the actual title. As suggestion, something like “The psychosocial determinants of Polish loneliness in the era of the COVID-19” could be more obviously informative and clear.
The proposal to change the title is interesting, but due to the territorially limited area of research, we cannot conclude that the research is representative for the entire Polish population.
Please, rewrite the abstract. Remove the sentences “1)Background:” “2) Methods:” “3) Results:” “4. Conclusions:”. It’s not necessary to relate herein the statistical results (e.g. rs =-0.54, p=0.0000). And look also for more keywords.
The summary, as suggested by the reviewer, has been revised. Quoting the numerical data of the research results was abandoned. Keywords added - quality of life and mental health.
The table 1 could be simpler. Too much information kills information.
Corrections were made to table 1 to slightly shorten the volume.
The title of Figure 1 is currently incomplete.
Figure 1 caption was supplemented with quality of life domains added.
Throughout the text, please reformulate the paragraphs. You make paragraphs with a single sentence. This is not at all desirable as it causes breaks in reading and interpretation. Review all the paragraphs.
The whole text has been corrected, the paragraphs have been corrected as suggested.
You should also reformulate the section “2.2. Research Methods”. In specific, from line 140 to 146.
Sections 2.2 were reformulated. as suggested.
Line 125: Google Forms is a online platform and not a software.
We agree with the suggestion, the text has been amended.
About the results, I am very frustrated. You only report descriptive data. Why don’t you test any t student test in order to clearly express and show any significant difference? By instance, with “no change” and “remote work” of data from table 6.
In this study, we focused on assessing factors related to loneliness. Our research was not aimed at a detailed analysis of the impact of remote work on the functioning of people during a pandemic. In future research, we would like to extend the analysis to look for a relationship between other variables related to pandemic constraints.
The Discussion section is a mess in its own right. Why reintroduce literature again instead of going straight to the key points in which the results obtained from previous studies are discussed?
The discussions were revisited and the suggested changes were made. Descriptive fragments have been removed.
Arguments from line 338 and 339 are confused.
We agree with the suggestion. Suggested arguments have been removed.
Line 357: you claim “my own research”, why? It should be “this study shows…”
The text was corrected as suggested.
In the “conclusions”, you managed to make 3 chapters with 4 sentences. Also rewrite the future research part, and add a chapter on limitations.
Conclusions have been completed. The limitations study was separated and the direction of future research was added.
Round 2
Reviewer 1 Report
Thank you. You made the necessary changes as suggested.
Author Response
Dear Reviewer, thank you very much for your positive opinion.
Reviewer 2 Report
Dear authors, here are some comments to improve your paper.
kind regards
Some observations from the literature can be included in introduction instead of discussion (e.g. anxiety and depression, which are not considered variables in the study)
In the description of the tests, you can report Crombach's alpha of your sample.
How are 'Experiencing negative emotions' and 'Lack of emotional control, aggressive behavior' assessed?
'Experiencing negative emotions' and 'Lack of emotional control, aggressive behavior' were assessed on the Likert scale (1-5). In the description of the methodology, these variables are described generally as the subjective assessment of mental health.
The variables need to be better described. They are not named and it is not known with which and how many questions they were evaluated.
The age of the sample is very large: from 18 to 84 years. By comparing more homogeneous age groups (perhaps by increasing the sample) the statistical magnitude might increase or other significant data might emerge.
We agree with the Reviewer's opinion. In the conducted analyzes, we focused only on age, as one of the many variables potentially influencing the level of loneliness. In the future, we would like to broaden the analysis by age group.
I do not see the comparison (Mann-Whitney U-test and/or the Kruskal-Wallis U-test) between different age groups. (E.g. "Various studies indicate a complex, curvilinear relationships occurring between the phases of life and loneliness. Loneliness is high among the youngest respondents, followed by a period of decreased loneliness in the middle age, rising again among the elderly".
Perlman, D. , (1990, 10-14 August). Age differences in loneliness: A meta-analysis. Paper presented at the 98th Annual Convention of the American Psychological Association, Boston, MA.
Perlman, D., Landolt, M. A. (1999). Examination of loneliness in children–adolescents and in adults: two solitudes or unified enterprise? In Rotenberg, K. J., Hymel, S. (Eds.), Loneliness in childhood and adolescence (pp. 325–347). Cambridge, England: Cambridge University Press)
Although the significance is high, the magnitude is low (see table 5)
Perhaps several variables can be assessed together with respect to the feeling of loneliness. E.g. a group in which both "The impact of the pandemic on the reduction of work" and "Lack of emotional control, aggressive behaviour" are high and a group in which both variables are low
334- 338 Literature reports indicate that women are significantly more likely than men to show tendencies to the occurrence of anxiety, depressive disorders and experiencing negative emotions [44-48]. Other researchers also argue that the experience of loneliness during the COVID-19 pandemic is associated with several risk factors, including those related to female sex.
In your research, no difference emerges between the gender variable and the feeling of loneliness or other variables. The data can be commented on.
In general, it may also be interesting to comment on or interpret some data where there are no significant differences, or which do not agree with what has emerged in the literature (E.g. table 6) The point is to highlight the data from this research.
Author Response
Dear Reviewer,
Thank you very much for your valuable comments and tips on our manuscript. We have made every effort to follow the recommendations and make the necessary corrections. We hope that our manuscript will meet editorial expectations in its current version.
